# An Anti-PSMA Immunotoxin Reduces Mcl-1 and Bcl2A1 and Specifically Induces in Combination with the BAD-Like BH3 Mimetic ABT-737 Apoptosis in Prostate Cancer Cells

**DOI:** 10.3390/cancers12061648

**Published:** 2020-06-22

**Authors:** Anie P. Masilamani, Viviane Dettmer-Monaco, Gianni Monaco, Toni Cathomen, Irina Kuckuck, Susanne Schultze-Seemann, Nathalie Huber, Philipp Wolf

**Affiliations:** 1Department of Urology, Medical Center—University of Freiburg, 79106 Freiburg, Germany; anie.priscilla.masilamani@uniklinik-freiburg.de (A.P.M.); irina.kuckuck@uniklinik-freiburg.de (I.K.); susanne.schultze-seemann@uniklinik-freiburg.de (S.S.-S.); nathalie.huber@uniklinik-freiburg.de (N.H.); 2Faculty of Medicine, University of Freiburg, 79106 Freiburg, Germany; viviane.dettmer@uniklinik-freiburg.de (V.D.-M.); mongianni1@gmail.com (G.M.); toni.cathomen@uniklinik-freiburg.de (T.C.); 3Institute for Transfusion Medicine and Gene Therapy, Medical Center—University of Freiburg, 79106 Freiburg, Germany; 4Center for Chronic Immunodeficiency, University of Freiburg, 79106 Freiburg, Germany

**Keywords:** prostate cancer, PSMA, targeted therapy, combination therapy, immunotoxin, ABT-737, apoptosis, Bcl-2 proteins

## Abstract

*Background:* Upregulation of anti-apoptotic Bcl-2 proteins in advanced prostate cancer leads to therapeutic resistance by prevention of cell death. New therapeutic approaches aim to target the Bcl-2 proteins for the restoration of apoptosis. *Methods:* The immunotoxin hD7-1(VL-VH)-PE40 specifically binds to the prostate specific membrane antigen (PSMA) on prostate cancer cells and inhibits protein biosynthesis. It was tested with respect to its effects on the expression of anti-apoptotic Bcl-2 proteins. Combination with the BAD-like mimetic ABT-737 was examined on prostate cancer cells and 3D spheroids and in view of tumor growth and survival in the prostate cancer SCID mouse xenograft model. *Results:* The immunotoxin led to a specific inhibition of Mcl-1 and Bcl2A1 expression in PSMA expressing target cells. Its combination with ABT-737, which inhibits Bcl-2, Bcl-xl, and Bcl-w, led to an induction of the intrinsic apoptotic pathway and to a synergistic cytotoxicity in prostate cancer cells and 3D spheroids. Furthermore, combination therapy led to a significantly prolonged survival of mice bearing prostate cancer xenografts based on an inhibition of tumor growth. *Conclusion:* The combination therapy of anti-PSMA immunotoxin plus ABT-737 represents the first tumor-specific therapeutic approach on the level of Bcl-2 proteins for the induction of apoptosis in prostate cancer.

## 1. Introduction

Prostate cancer (PC) represents the second most common cancer in men with more than 1.2 million new cases and more than 350,000 deaths estimated worldwide [1]. Despite improved therapeutic options, such as androgen deprivation therapy, radiation, and chemotherapy, metastatic PC remains incurable. The reason is that PC is driven by alterations in several signaling pathways promoting tumor progression and resistance followed by treatment failure [2].

One of these signaling pathways is the intrinsic apoptotic pathway. Therapeutic intervention including ADT, radiation, and chemotherapy aims to invoke sufficient cytotoxic stress to initiate a pro-apoptotic signaling and critically depends on the integrity of this pathway to actually execute cell death. In living cells, pro-apoptotic proteins (Bax, Bak) are bound by anti-apoptotic proteins (Bcl-2, Bcl-xl, Mcl-1, Bcl-w, Bcl2A1) of the Bcl-2 family. After cell stress or damage, the intrinsic apoptotic pathway is activated. The pro-apoptotic activators BID, BIM, and PUMA and the sensitizers BAD and NOXA are upregulated and bind via their so-called BH3 (B-cell lymphoma-2 (Bcl-2) homology domain 3) domain to the anti-apoptotic proteins. NOXA preferentially sequesters Mcl-1 and Bcl2A1, whereas BAD binds Bcl-2, Bcl-xl, and Bcl-w. Then Bax and Bak are released to form pores in the outer membrane of mitochondria, an event, which is termed mitochondrial outer membrane permeabilization (MOMP). MOMP is followed by cytochrome C release into the cytosol, activation of effector caspases, and finally cell death (reviewed in [3,4]).

Cancer cells are marked by an upregulation of the anti-apoptotic Bcl-2 proteins increasing the threshold for triggering apoptosis and leading to therapeutic resistance [5,6,7,8]. Whereas overexpression of Bcl-2 seems to be a main characteristic of leukemia cells, upregulation of Bcl-xl, Mcl-1, Bcl-w, and Bcl2A1 is preferentially found in solid tumors, including PC [7,9,10,11,12,13,14,15]. Interestingly, the pro-apoptotic members Bax and Bak are omnipresent in all stages of PC, and mutations of both proteins that could affect their function were found to be only rare events (reviewed in [12,16]). Therefore, neutralization of the anti-apoptotic Bcl-2 proteins for the activation of the intrinsic apoptotic pathway represents a promising therapeutic option for the treatment of advanced PC. 

Downregulation of anti-apoptotic Bcl-2 proteins can be done using DNA-antisense and RNA interference methods. A combination of such methods with chemotherapy or radiation against PC elicited synergistic effects in preclinical studies [17,18,19,20]. However, in a phase II study with the Bcl-2 antisense nucleotide oblimersen sodium (Genasense^®^, Genta Inc., Berkeley Heights, NJ, USA) in combination with docetaxel the primary endpoints (PSA response > 30% and major toxicity event rate < 45%) were not reached [21]. As alternatives, Bcl-2 antagonists, called BH3 mimetics, were generated, which can bind to the anti-apoptotic Bcl-2 proteins to release Bax and Bak (reviewed in [22]). One prominent *pan*-BH3 mimetic, called Gossypol (AT-101, Ascenta Therapeutics Inc., Malvern, PA, USA), inhibits Bcl-2, Bcl-xl, and Bcl-w and, to a lesser extent, Mcl-1. It was tested in a phase I/II trial in patients with metastatic castration-resistant PC (mCRPC). However, only two of 23 patients experienced a PSA decrease without objective responses [23]. Moreover, combination of AT-101 with different chemotherapeutic agents only showed modest clinical efficacy and sometimes dose-limiting toxicities [24,25]. Therapeutic failure of *pan*-BH3 mimetics can be traced back to their lack of tumor specificity and their low affinity to some of the anti-apoptotic Bcl-2 proteins. 

Since *pan*-BH3 mimetics were not specific enough to have therapeutic effects, specific BH3 mimetics were generated. ABT-737 (4-[4-[[2-(4-chlorophenyl)phenyl]methyl]piperazin-1-yl]-N-[4-[[(2R)-4-(dimethylamino)-1-phenylsulfanylbutan-2-yl]amino]-3-nitrophenyl]sulfonylbenzamide) and its orally available analogue ABT-263 (Navitoclax) specifically bind Bcl-2, Bcl-xl, and to a lesser extent Bcl-w [26]. They thus take over the function of BAD and are therefore also called BAD-like proteins [26]. ABT-737/263 monotherapy was not efficient for the induction of apoptosis in PC cells, because Mcl-1 and Bcl2A1 are not inhibited. After combination of ABT-737/ABT-263 with chemotherapeutic agents or inhibitors that additionally target Mcl-1, synergistic cytotoxic effects were measured. None of the tested combination therapies, however, have so far been tumor-specific, so they are therefore expected to cause adverse side effects in the clinic that could outweigh their clinical benefits (reviewed in [22]).

In a first study, we developed a new therapeutic approach joining tumor specificity with a direct induction of apoptosis on the level of the Bcl-2 proteins. For this, we generated an immunotoxin, called D7(VL-VH)-PE40. It consists of the single chain variable fragment (scFv) D7 from the monoclonal anti-PSMA antibody 3/F11 in VL-VH orientation binding to the prostate specific membrane antigen (PSMA). PSMA is a glycoprotein on the surface of prostate tumor cells with enhanced expression in advanced stages, including castration-resistant PC (CRPC) [27,28,29]. Different anti-PSMA antibodies and small molecules have been developed in the past for targeted diagnosis and therapy of PC [30,31,32,33]. The second domain of our immunotoxin is PE40, the toxin domain of *Pseudomonas aeruginosa* Exotoxin A (PEA) [34]. PEA is a virulence factor with enzymatic activity leading to ADP-ribosylation of the eukaryotic elongation factor 2 (eEF-2) on ribosomes. This is followed by an inhibition of protein biosynthesis and induction of apoptosis (reviewed in [35]). Combination of our immunotoxin D7(VL-VH)-PE40 at low concentrations with ABT-737 elicited synergistic cytotoxicity in PSMA-expressing PC cells [36]. 

In the present study, we tested a derivate, called hD7-1(VL-VH)-PE40, containing a humanized scFv in combination with ABT-737 to further characterize the targeted induction of apoptosis. We found that the immunotoxin led to a reduction of Mcl-1 and Bcl2A1. It thus undertakes the function of NOXA, namely, to reduce free Mcl-1 and Bcl-1A for binding to Bax and Bak in order to induce apoptosis. In combination with ABT-737, a specific synergistic cytotoxicity based on apoptosis in PSMA-expressing PC cells and 3D spheroids was found. Moreover, a significantly prolonged survival was reached with the immunotoxin/ABT-737 combination therapy in a PC SCID mouse xenograft model. 

## 2. Results

The immunotoxin hD7-1(VL-VH)-PE40 was generated by cloning the anti-PSMA scFv hD7-1 via NcoI/NotI restriction sites into the plasmid pHOG21. The cytotoxic domains II, Ib, and III from *Pseudomonas* Exotoxin A (PE40) were C-terminally cloned to the scFv using the XbaI restriction site. The immunotoxin includes a human c-myc tag for detection and a His_6_ tag for purification (Figure 1a). The immunotoxin was expressed in *E.coli* XL-1 blue bacteria and successfully obtained in high purity of about 82% after IMAC (Figure 1b). Western blot analysis verified the expression of the 70 kDa immunotoxin (Figure 1c). Specific binding of the immunotoxin was measured on the PSMA-positive PC cell lines LNCaP and C4-2 by flow cytometry. No binding was seen to PSMA-negative PC-3 cells (Figure 1d). 

Inhibition of protein biosynthesis was tested by puromycin-based Western blot. The antibiotic puromycin can enter the ribosomal A-site of eukaryotic cells and causes translation termination by binding to newly synthesized proteins. These proteins can be detected by Western blot analysis with help of an anti-puromycin antibody. As shown in Figure 2a, the immunotoxin hD7-1(VL-VH)-PE40 completely inhibited the protein biosynthesis of LNCaP and C4-2 cells after 24 h, but not that of PC3 control cells. Inhibition of protein biosynthesis was accompanied by a reduction of cell viability. The immunotoxin specifically reduced the viability of LNCaP and C4-2 cells with IC_50_ values of 130 pM and 227 pM after 48 h (Figure 2b). 

Next, we examined the induction of MOMP in the PC cells treated with hD7-1(VL-VH)-PE40 with help of the Rhodamine-123 assay. The cationic dye Rho-123 congregates within the negatively charged mitochondrial matrix. An intact membrane results in dye accumulation and a positive fluorescence signal. If the outer membrane loses its integrity during MOMP, the dye will no longer associate with the mitochondria, resulting in a loss of fluorescence intensity. As shown in Appendix A, a time- and concentration-dependent induction of MOMP was detected in LNCaP and C4-2 cells 4–48 h after incubation with the immunotoxin. Percentage of immunotoxin treated PC-3 control cells undergoing MOMP did not differ from that of the untreated control, underlining the high specificity of the immunotoxin.

Expression of the apoptosis-related proteins was examined by Western Blot analysis in cell lysates 4, 24, and 48 h after incubation with 1 nM hD7-1(VL-VH)-PE40 (Figure 3). The immunotoxin markedly reduced the expression of the anti-apoptotic proteins Mcl-1 and Bcl2A1 in LNCaP and C4-2 cells after 24 and 48 h, whereas Bcl-2, Bcl-xl, and Bcl-w levels remained stable. No or only slight differences were observed in the expression of the pro-apoptotic proteins Bax and Bak during treatment. Activation of the effector caspase-3 and PARP cleavage were detected after 24 h in LNCaP lysates and after 48 h in C4-2 lysates and pointed to the execution of the intrinsic apoptotic pathway. A lack of caspase-8 cleavage showed that the extrinsic apoptosis pathway was not activated by the immunotoxin. No differences in the expression of apoptosis-related proteins were found in PC-3 cells after immunotoxin treatment.

Since hD7-1(VL-VH)-PE40 downregulated Mcl-1 and Bcl2A1, we combined the immunotoxin with ABT-737, which preferentially targets Bcl-2, Bcl-xl, and Bcl-w, to effectively overcome the threshold for triggering apoptosis. With the inhibitor ABT-737 alone IC_50_ values of 5.46 µM, 6.17 µM and 16.94 µM were reached on LNCaP, C4-2 and PC3 cells after 48 h, respectively (Appendix A). We used low doses of 0.63 µM ABT-737 and 40-160 pM hD7-1(VL-VH)-PE40. For the following combination experiments. 86.9% and 86.8% viable LNCaP cells were found after 48 h incubation with ABT-737 or immunotoxin alone, respectively (Figure 4a). On C4-2 cells, a viability of 114.6% and 69.9% was determined after treatment (Figure 4b). In combination, however, both agents significantly reduced the viability of LNCaP cells to 35.0% (Figure 4a) and the viability of C4-2 cells to 21.3% (Figure 4b). Based on the Bliss Independence Model according to Zhao et al., the cytotoxicity of the immunotoxin in combination with ABT-737 was determined to be synergistic on both cell lines [37]. In contrast, PSMA-negative PC-3 cells remained unaffected by combination treatment (*p* > 0.05, Figure 4c). Caspase-3 and PARP cleavage confirmed that the synergism was based on the induction of apoptosis (Figure 4d). 

Next, we tested whether the combination was also effective for the treatment of C4-2 3D spheroids. As shown in Figure 5a, spheroids were formed in 3D CoSeedis™ wells for 3 days, before they were spun down into 6-well plates. During treatment for 96 h, spheroids formed aggregates. After 96 h, viability of the spheroids was significantly reduced by immunotoxin treatment compared to the control (*p* = 0.0118). Combination treatment induced significant reduction of cell viability compared to the control (*p* < 0.0001) and monotherapies (*p* = 0.0003 for combination vs. ABT-737, *p* = 0.0002 for combination vs. immunotoxin; Figure 5a,b). Its high cytotoxicity on 3D spheroids led us to test the combination therapy in the C4-2 SCID mouse xenograft model.

For testing the in vivo toxicity of ABT-737, mice were treated with 25, 50, or 100 mg/kg bw ABT-737. Control mice received the dilution buffer as vehicle. No external signs of acute toxicity were observed for any of the animals over a period of 7 days, and no changes in body weight were measured in any of the treatment groups (Appendix A). Our observations are in line with previous studies that ABT-737 can be administered safely as a single dose or repeatedly in mice without causing acute toxicity, such as hepatotoxicity [38,39]. Since thrombocytopenia is the main adverse side effect in the clinic, we investigated in the next step the effect of the BH3 mimetic on the blood count to find suitable concentrations for the therapy trial. As shown in Appendix A, ABT-737 induced transient thrombocytopenia in SCID mice at a concentration of 50 mg/kg bw. The number of thrombocytes had already decreased 3 h after treatment, but the initial number was reached again after 72 h. The transient thrombocytopenia was accompanied by slight leukocytosis, whereas the number of erythrocytes was not affected. External signs of acute toxicity were not observed in mice at any doses. We therefore decided to use 10 mg/kg bw ABT-737 for the first two injections on days 1 and 3 of treatment and 25 mg/kg bw for the third injection at day 8 to exclude adverse side effects from the BH3 mimetic during therapy (Figure 6a).

Animals from different treatment groups showed comparable average tumor sizes at the beginning of treatment (61.8 +/− 13.7 mm³ for the control group, 62.9 +/− 8.1 mm³ for the ABT-737 monotherapy group, 65.2 +/− 13.5 mm³ for the immunotoxin monotherapy group, 61.3 +/− 19.3 mm³ for the combination treatment group). Mice survived significantly longer when treated with combination therapy compared to the other groups (*p* = 0.012 for combination vs. control; *p* = 0.026 for combination vs. ABT-737; *p* = 0.043 for combination vs. immunotoxin). The median survival time was 28.1 days for the animals treated with combination compared to 15.6 days, 15.8 days, and 16.7 days for the control, ABT-737, and immunotoxin groups, respectively (Figure 6b). Enhanced survival of mice treated with combination therapy was based on a significant inhibition of tumor growth, as demonstrated by the calculation of the time-adjusted area under the curve (AUC, *p* > 0.0001 for combination vs. control; *p* = 0.0008 for combination vs. ABT-737; *p* = 0.0497 for combination vs. immunotoxin; Figure 6c).

## 3. Discussion

The survival of prostate tumor cells and their resistance against conventional therapies mainly depend on their resistance to apoptosis. BH3 mimetics that inhibit the anti-apoptotic Bcl-2 proteins are suitable for effective intervention in the intrinsic apoptotic pathway. However, they were found not to be effective in clinical studies against PC (reviewed in [22]). This can be explained by structural differences between Bcl-2 proteins, so that BH3 mimetics cannot inhibit all of them or bind unspecifically. Moreover, especially Bcl-2, Bcl-xl, and Bcl-w are known to be stable proteins with long half-lives, which may require continued inhibition [3,40]. Thus, it remains questionable whether BH3 mimetics can be successfully applied as monotherapy in cancer patients in a sufficient dosage without dose-limiting side effects. 

Our work confirms that the specific BH3 mimetic ABT-737 has only low cytotoxicity in the cell lines LNCaP, C4-2 and PC-3, which represent androgen-dependent and independent stages of metastatic PC and show high expression of anti-apoptotic Bcl-2 proteins. Compared to ABT-737, a specific and about 22,000–40,000-fold higher cytotoxicity was reached with hD7-1(VL-VH)-PE40. This can be attributed to the enzymatic activity of our immunotoxin, which allows one molecule of immunotoxin to ADP-ribosylate several eEF-2 molecules, and thus effectively inhibit protein biosynthesis. In contrast, one molecule ABT-737 can only inhibit one anti-apoptotic protein and therefore does not act as comprehensively as the immunotoxin.

In the past, PE-based immunotoxins against different tumor entities were shown to downregulate Mcl-1 [41,42,43]. We could show that our immunotoxin hD7-1(VL-VH)-PE40 not only downregulated Mcl-1, but additionally Bcl2A1. Both are proteins with short half-lives from a few minutes to a few hours based on constitutive protein turnover through poly ubiquitination and proteasomal degradation [44,45]. Our immunotoxin is thus comparable to the apoptosis inducer NOXA in that it reduces the amount of free Mcl-1 and Bcl2A1 for the release of Bax and Bak (Figure 7). It is, therefore, plausible that our immunotoxin showed a synergistic cytotoxicity with the BAD-like mimetic ABT-737 for the effective induction of the intrinsic apoptotic pathway. 

Our study is also in line with previous work, in which Mcl-1 and Bcl2A1 overexpression was associated with ABT-737 resistance [46,47,48]. Based on that, NOXA activation was suggested as a new strategy for combination with ABT-737 and against ABT-737 resistant cells. For example, the vinca alkaloid vinblastine as well as the proteasome inhibitors celastrol and MLN2238 were found to upregulate NOXA and sensitize tumor cells to ABT-737 [49,50,51]. In contrast to our study, however, in which an immunotoxin against PSMA was used, these substances are not tumor specific. 

The combination of hD7-1(VL-VH)-PE40 with ABT-737 was not only effective in vitro against monolayer cell culture and 3D spheroids, but also in vivo in the C4-2 SCID mouse xenograft model. We used immunotoxin doses shown to be non-toxic in this mouse model [34] and ABT-737 doses which did not elicit thrombocytopenia, which is known to be the most adverse side effect in patients due to platelet dependence on Bcl-xl [52]. The combination resulted in a significantly prolonged survival of the animals based on an inhibition of tumor growth. To our knowledge, our study is the first one that introduces a targeted therapeutic concept against advanced PC for the induction of apoptosis based on the Bcl-2 protein level.

## 4. Materials and Methods

### 4.1. Cells and Chemicals

Identities of the of the PSMA-positive PC cell lines LNCaP and C4-2 and the PSMA-negative cell line PC-3 (ATCC, Manassas, VA, USA) were verified using short tandem repeat (STR) analysis (CLS GmbH, Eppelheim, Germany). LNCaP and C4-2 cells were propagated in RPMI 1640 medium and PC-3 cells in F12 medium (Gibco, Invitrogen, Karlsruhe, Germany) containing 10% fetal calf serum (Biochrom, Berlin, Germany) and penicillin/ streptomycin (100 U/mL, 100 mg/L) at 37 °C and 5% CO_2_. ABT-737 (Abcam, Cambridge, UK) was dissolved in DMSO as a stock solution of 20 mM and stored at 4 °C. For the animal experiments ABT-737 (0.5 g/mL in DMSO) was diluted in a mixture of 30% propylene glycol (Sigma Aldrich, Darmstadt, Germany), 5% Tween 80, 65 D5W (5% dextrose in water), pH 4.0 and sterile-filtered.

### 4.2. Cloning, Expression, and Purification of the Immunotoxin hD7-1(VL-VH)-PE40

The genes of the humanized variable domains of the heavy (VH) and light chains (VL) of the anti-PSMA scFv D7 including a Glycin-Serin rich linker, were synthesized by GeneArt (Regensburg, Germany) and cloned into the expression vector pHOG21 N-terminally to the cytotoxic domain of *Pseudomonas aeruginosa*, PE40 (Figure 1a). Cloning was verified by sequencing (Eurofins Genomics, Ebersberg, Germany). *E.coli* XL-1 blue bacteria (Agilent Technologies, Santa Clara, CA, USA) were chemically transformed with the vector, and the immunotoxin hD7-1(VL-VH)-PE40 was periplasmatically expressed and purified using immobilized metal affinity chromatography (IMAC) as described earlier [34]. In brief, bacteria were grown overnight in YT medium containing 0.1 M glucose + 100 mg/mL ampicillin, then diluted 1:20 and grown again as 600 mL cultures at 37 °C. When cultures reached an OD of 0.8, bacteria were pelleted by centrifugation and resuspended in the same volume of fresh medium containing 50 mg/mL ampicillin, 0.4 M sucrose, and 1 mM IPTG. After incubation for 18 h at RT, bacteria were harvested by centrifugation and resuspended in 30 mL of ice-cold 50-mM Tris-HCl, 20% sucrose, 1-mM EDTA (pH 8.0) to isolate soluble periplasmic proteins. After incubation for 1 h on ice, the spheroblasts were centrifuged yielding a soluble periplasmic extract (PPE) in the supernatant that was dialyzed against 50 mM Tris-HCl, 1 M NaCl (pH 7.0). Purification by IMAC was done using a 1 mL column with chelating Sepharose (Amersham Biosciences, Freiburg, Germany) charged with Ni^2+^ and equilibrated with a buffer containing 50 mM Tris-HCl and 1 M NaCl (pH 7.0). The PPE was loaded onto the column, washed with 20 column volumes of equilibration buffer containing 40 mM imidazole (wash fraction) and then eluted with 3 × 1.5 mL of the same buffer containing 250 mM imidazole. Elution fractions were dialyzed against phosphate-buffered saline (PBS). Determination of the protein content was quantified by Nanodrop analysis (Thermo Fisher Scientific, Waltham, MA, USA). Purity was calculated by densiometric analysis of the SDS gel using Image Lab™ software (Bio-Rad Laboratories, Feldkirch, Germany). The immunotoxin was aliquoted and stored at −20 °C.

### 4.3. SDS-PAGE and Western Blot Analysis

Expression and purity of the immunotoxin preparations as well as the expression of apoptotic proteins was examined by SDS-PAGE and Western blot analysis, as described before [34]. For detection of the immunotoxin mouse anti-human c-myc mAb (Roche Diagnostics, Mannheim, Germany, # 11814150001) was used. The following antibodies were used for the detection of the apoptotic proteins (50 µg per lane): Bcl-2 mouse mAb (Thermo Fisher Scientific, Darmstadt, Germany, # 13-8800), Bcl-xl rabbit mAb (Cell Signaling Technology Europe, Leiden, The Netherlands, # 2764), Mcl-1 rabbit mAb (Cell Signaling Technology Europe, Frankfurt am Main, Germany # 5453), Bcl-w rabbit mAb (Thermo Fisher Scientific, Darmstadt, Germany # PA5-51430), Bcl2A1 rabbit pAb (Merck Chemicals, Darmstadt, Germany, # ABC498), Bax rabbit pAb (Santa Cruz Biotechnology, Heidelberg, Germany, # sc-493), Bak rabbit mAb (Cell Signaling Technology Europe, # 6947), Cas-3 mouse mAb (ECM Biosciences, Versailles, KY, USA, # CM4911), Cas-8 mouse mAb (Cell Signaling Technology Europe, # 9746) PARP rabbit pAb (Cell Signaling Technology Europe, # 9542) ß-actin rabbit pAb (Cell Signaling Technology Europe, # 5125). HRP conjugated rabbit anti-mouse pAb (Dako, Hamburg, Germany, # P0161) and HRP conjugated goat anti-rabbit pAb (LI-COR Biosciences, Bad Homburg, Germany, # 26-80011) were used as secondary antibodies. Western blots were developed with an enhanced chemiluminescence (ECL) system and protein bands were detected and analyzed with help of a ChemoDoc™ MP Imaging System and the software Image Lab™ (Appendix A).

### 4.4. Inhibition of Protein Biosynthesis

For puromycin Western blot, target cells were incubated with the immunotoxin hD7-1(VL-VH)-PE40 for 0, 4, 24, and 48 h. Before harvesting, cells were treated with 5 µg/mL puromycin (Tocris #4089, Bio-Techne GmbH, Wiesbaden, Germany) for 15 min. Cells were lysed with RIPA buffer (50 mM Tris-HCl, 150 mM NaCl, 1 mM EDTA, 0.5% NaDeoxycholate, 0.05% SDS, 1% Igepal). Anti-puromycin mouse mAb (Merck, # MABE343) and HRP conjugated rabbit anti-mouse Ab (Dako, #P0161) were used for the detection of proteins by Western Blot that were translated at the time of lysate preparation.

### 4.5. Flow Cytometry

Binding of the immunotoxin hD7-1(VL-VH)-PE40 to cell-adherent PSMA was measured by flow cytometry as described previously [34]. Mouse anti-human c-myc mAb (BD Biosciences, Heidelberg, Germany, # 51-1485GR) and goat anti-mouse Ig-R-PE Ab (Southern Biotech, Birmingham, AL, USA, # 1010-09) were used for the detection of bound immunotoxin.

### 4.6. Rhodamine 123 Assay

For the detection of MOMP, the Rhodamine 123 (Rho-123) assay was performed. For this, 1.5 × 10^5^ cells/well were seeded in 6-well plates and grown overnight in 5% CO_2_ at 37 °C. Then cells were treated with the immunotoxin hD7-1(VL-VH)-PE40 or ABT-737 for 4, 24, or 48 h. 4 mg/mL cycloheximide was used as positive control. After that cells were trypsinized and centrifuged (430 g, 3 min). The pellet was resuspended in 10 µL PBS + 3% FCS + 0.1% sodium azide and incubated with 25 µL rhodamine 123 (10 µg/mL, Merck). After incubation for 30 min at 37 °C, cells were washed three times with PBS and resuspended in 200 µl PBS + 1 µg/mL propidium jodide (Merck). Decreasing fluorescence intensity of rhodamine 123 as a sign for MOMP was detected using a FACSCalibur flow cytometer and the software CellQuest Pro (BD Biosciences).

### 4.7. WST-1 Cell Viability Test

The WST-1 assay (Roche Diagnostics) was used to examine the effects of the immunotoxin and ABT-737 alone or in combination on the cultured cell lines as described before [34]. The IC_50_ values, defined as the immunotoxin/ABT-737 concentrations leading to a reduction of 50% cell viability, were estimated by non-linear regression [log (inhibitor) vs. response (three parameters)] using software GraphPad Prism 7.

### 4.8. Cytotoxicity on 3D Spheroids

3D CoSeedis™ (abc biopply, Cham, Switzerland) containing 200 microwells were placed on 6-well plates and equilibrated with 5 mL of cell culture medium for 3 h. After equilibration, 10^6^ C4-2 cells in 7 mL medium were seeded into the microwells. After 48 h CoSeedies™ were cut into four equal pieces and spheroids from each piece were spun down (300 g, 2 min) into 6-well plates containing 2 mL fresh medium. Then spheroids were treated with hD7-1(VL-VH)-PE40 or ABT-737 alone or in combination. After 96 h spheroid cells were trypsinized, collected and stained with 0.1% trypan blue solution. Cell viability of was determined using a hemocytometer.

### 4.9. In Vivo Experiments

All animal experiments were carried out according to the German animal protection law with permission from the responsible local authorities (G-16/90 from 07.07.2016, RP Freiburg). Male SCID CB17/lcr-Prkdc scid/Crl mice (5–6 weeks old, 20–25 g) were purchased from Janvier Labs (Saint-Berthevin, France) and kept under sterile and standardized environmental conditions. For testing toxicity of ABT-737, groups of three animals each were treated with single doses of 25, 50, or 100 mg/kg bw ABT-737. Control mice received the dilution buffer as vehicle. Mice were observed for apparent signs of toxicity (loss of weight and appetite, changes in pelage, fever, tension, apathy, aggression, respiratory disorders, paralyses, death) over a period of 7 days.

Since thrombocytopenia is known to be the most adverse side effect of ABT-737, 3–10 animals per group were injected i.p. with different doses of ABT-737 and blood was taken with help of a CB 300 K2E microvette (Sarstedt, Nümbrecht, Germany) 0, 3, 6, 24, and 72 h after treatment. Afterwards the blood cells were counted (scil Vet ABC™Animal Blood Counter, SACC, Gurnee, IL, USA).

For therapy, SCID mice were subcutaneously injected with 2.5 × 10^6^ C4-2 cells in 100 μL PBS mixed with 100 μL Matrigel into the right flank (day 1 of treatment). Growing tumors were palpated and tumor diameters were measured in two axes using a caliper. Tumor volumes were calculated using the formula V = (d² × D)/2, where d was the smaller diameter and D the larger diameter of the tumor. When tumors reached a mean tumor volume of about 60 mm^3^ [30–90 mm³], mice were randomized into four groups (n = 6–8). The first group received combinatorial treatment with 0.15 mg/kg bw hD7-1(VL-VH)-PE40 i.v. and 10 mg/kg bw ABT-737 i.p. one hour later at days 1 and 3, followed by 0.15 mg/kg bw hD7-1(VL-VH)-PE40 i.v. and 25 mg/kg bw ABT-737 i.p at day 8. The second group was injected with the immunotoxin alone and the third one with ABT-737 alone. The control group remained untreated. During the experiment, tumor sizes and body weights of the animals were measured 2–3 times a week. Mice were euthanized when one or more of the following termination criteria were met: tumor diameter >15 mm, invasive tumors, ulcerating tumors, or weight loss >20%, body condition score (BCS) >3, spasms, paralysis, body curvature respiratory disorders, apathy, or aggressiveness.

### 4.10. Statistics

Significance of the data from WST-test, trypan blue assay, and blood analysis was calculated using the Student’s *t* test (unpaired, parametric with Welch’s correlation). The Bliss Independence Model according to Zhao et al. was used to determine synergistic cytotoxicity of the immunotoxin in combination with ABT-737 on the target cells [37]. Kaplan-Meier analysis and log-rank test were used for the comparison of animal survival (Software GraphPad Prism 8.0.0 for Windows, GraphPad Software, San Diego, CA, USA). Tumor growth inhibition was determined by calculating the time-adjusted Area Under the Curve (AUC) (Software GraphPad Prism version 8.2.1, San Diego, CA, USA), as recommended by Duan et al. [53]. Values between measurements were interpolated using the “natural” method of the spline function (https://www.R-project.org/). Differences in AUC were analyzed using the Student’s *t* test (unpaired, parametric with Welch’s correlation).

## 5. Conclusions

Induction of cell death is a ‘conditio sine qua non’ in any anticancer strategy. The combination of hD7-1(VL-VH)-PE40 with ABT-737 leads to a comprehensive activation of the intrinsic apoptotic pathway in a comparable way that occurs when the sensitizer proteins NOXA and BAD are activated for the triggering of cell death. It is largely tumor-specific, acts independently of acquired mutations in upstream signaling pathways that lead to apoptosis resistance, and can thus overcome this resistance in PC cells. Our combination therapy could reduce dose-limiting side effects in future clinical trials and achieve anti-tumor efficacy in patients with advanced PC.

## Figures and Tables

**Figure 1 cancers-12-01648-f001:**
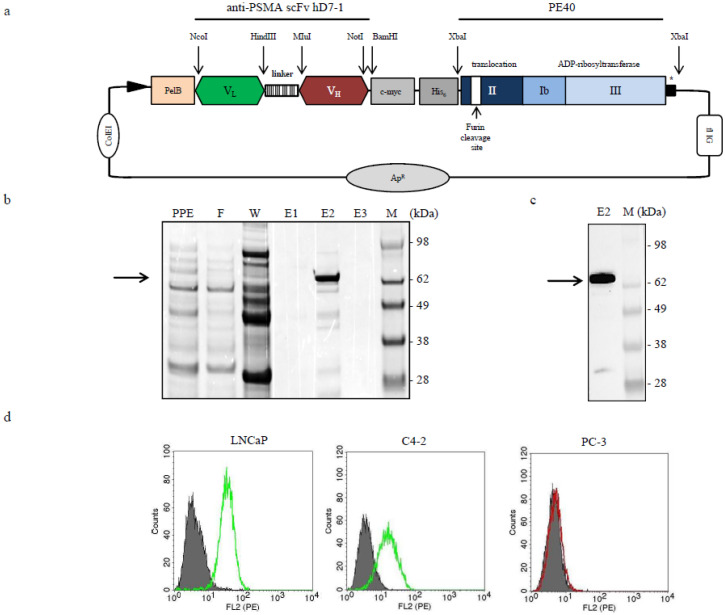
Generation and in vitro characterization of the anti-PSMA immunotoxin hD7-1(VL-VH)-PE40. (**a**) Schematic representation of the anti-PSMA immunotoxin hD7-1(VL-VH)-PE40 cloned in the vector pHOG21. (**b**) SDS-PAGE and (**c**) Western blot analysis of the purified immunotoxin (arrow), which was found in the elution fraction E2. (**d**) Binding of the immunotoxin at saturation concentration to PSMA-positive LNCaP and C4-2 cells and PSMA-negative PC-3 cells as shown by flow cytometry. Histograms with mouse anti-human c-myc mAb and goat anti-mouse Ig-R-PE alone are shown in grey. Abbreviations: *C-myc*, human c-myc tag; *His_6_*, hexahistidine tag; *E1-3*, elution fractions; *F*, flow through; *linker*, (Gly_4_Ser)_3_ linker; *M*, marker; *PPE*, periplasmatic extract; *pelB*, pel B leader; *V_L_*, variable domain of the antibody light chain; *V_H_*, variable domain of the antibody heavy chain; *W*, wash fraction.

**Figure 2 cancers-12-01648-f002:**
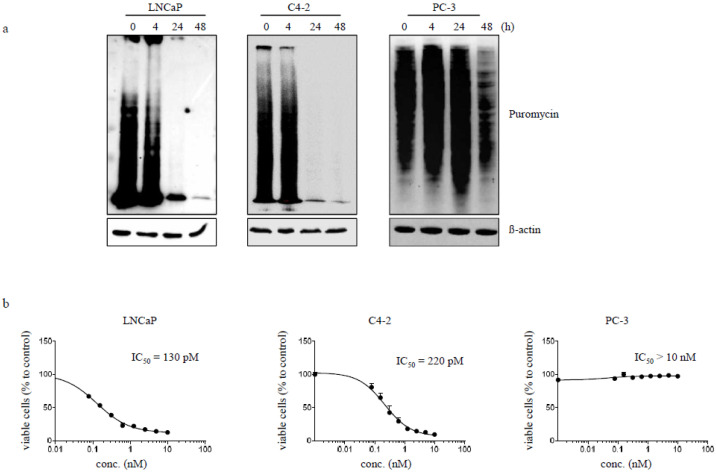
Influence of the anti-PSMA immunotoxin hD7-1(VL-VH)-PE40 on protein biosynthesis and viability of PC cells. (**a**) Time-dependent inhibition of protein biosynthesis in LNCaP, C4-2 and PC-3 cells after incubation with 1 nM immunotoxin as demonstrated by puromycin Western blot analysis. (**b**) PSMA-positive LNCaP and C4-2 cells as well as PSMA-negative PC-3 cells (control) were incubated with the immunotoxin for 48 h. Reduction of cell viability was calculated using WST-1 test. Mean values ± SD of three independent experiments.

**Figure 3 cancers-12-01648-f003:**
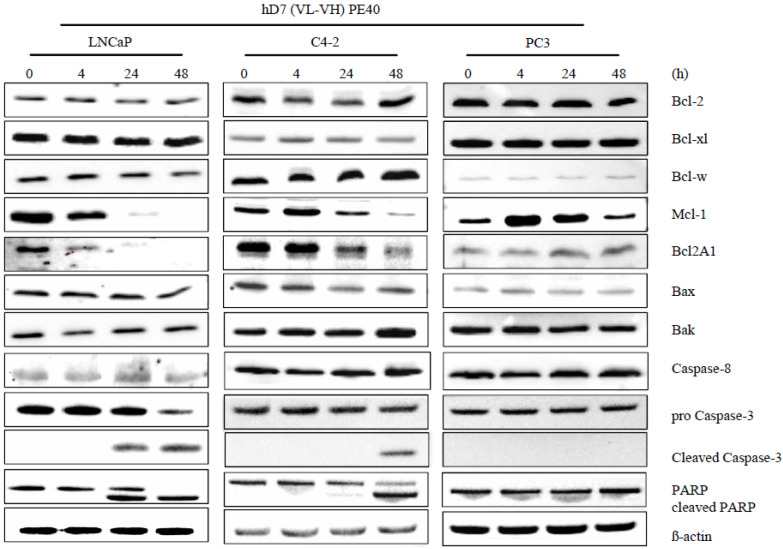
The immunotoxin hD7-1(VL-VH)-PE40 induces apoptosis in PSMA-positive cells by reducing the expression of Mcl-1 and Bcl2A1. PC cells were incubated for different time points with 1 nM immunotoxin. After cell lysis, expression of the anti-apoptotic Bcl-2 proteins Bcl-2, Bcl-xl, Mcl-1, Bcl-w, and Bcl2A1, the pro-apoptotic Bcl-2 proteins Bax and Bak, as well as the activation of caspase-3, and -8, and the cleavage of PARP were determined by Western blot. ß-actin was detected as loading control.

**Figure 4 cancers-12-01648-f004:**
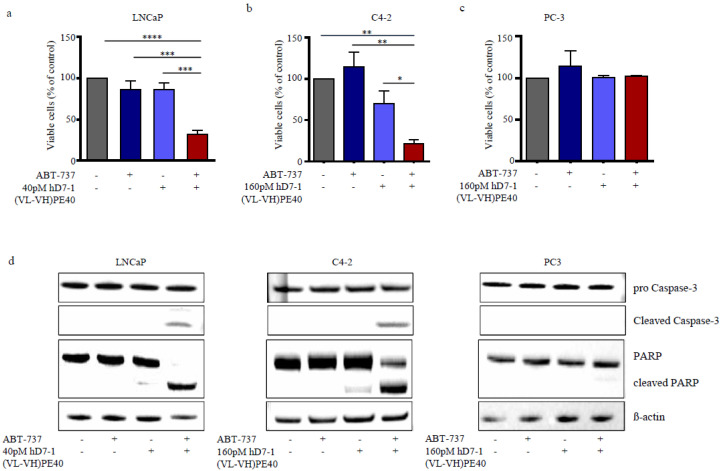
Combination of low doses immunotoxin hD7-1(VL-VH)-PE40 and ABT-737 elicit synergistic cytotoxicity based on apoptosis in PC cells. (**a**) PSMA-positive LNCaP and (**b**) C4-2 cells and (**c**) PSMA-negative PC-3 cells (control) were incubated with immunotoxin and ABT-737 alone or in combination for 48 h. Cytotoxicity was determined by WST-1 assay. Mean ± SD of three independent experiments. Statistically significant differences were determined with unpaired *t*-test: * *p* < 0.05, ** *p* < 0.01, *** *p* < 0.001, **** *p* < 0.0001. (**d**) Combination of hD7-1(VL-VH)-PE40 and ABT-737 led to Caspase-3 activation and PARP cleavage after 48 h, as shown by Western blot analysis.

**Figure 5 cancers-12-01648-f005:**
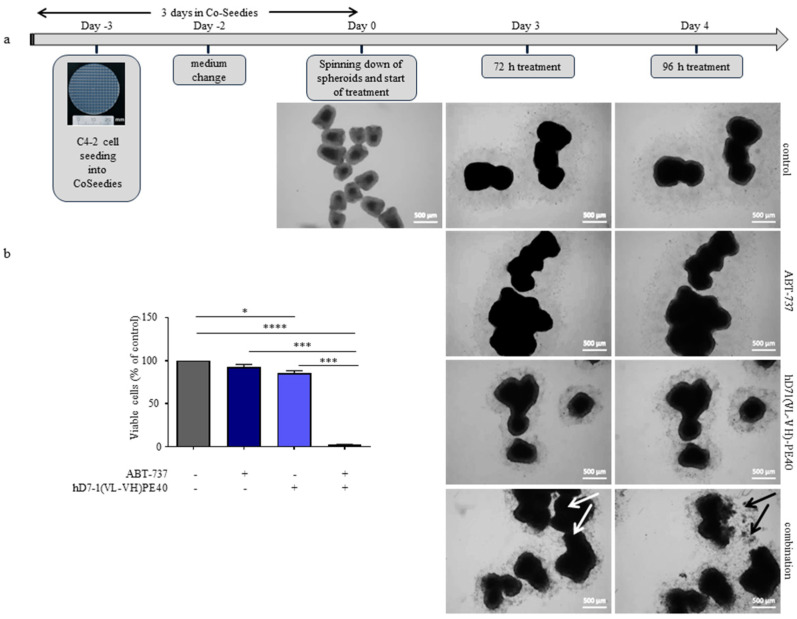
Combination of low doses immunotoxin hD7-1(VL-VH)-PE40 and ABT-737 induce synergistic cytotoxicity in 3D spheroids of C4-2 cells. (**a**) C4-2 spheroids were incubated with low doses immunotoxin and ABT-737 alone or in combination. Parts of the spheroids disintegrated in the combination treatment group between 72 and 96 h (white and black arrows). (**b**) Analysis of viable spheroid cells 96 h after treatment as determined by trypan blue assay. Mean ± SD with statistically significant differences determined by unpaired *t*-test: * *p* < 0.05, *** *p* < 0.001, and **** *p* < 0.0001.

**Figure 6 cancers-12-01648-f006:**
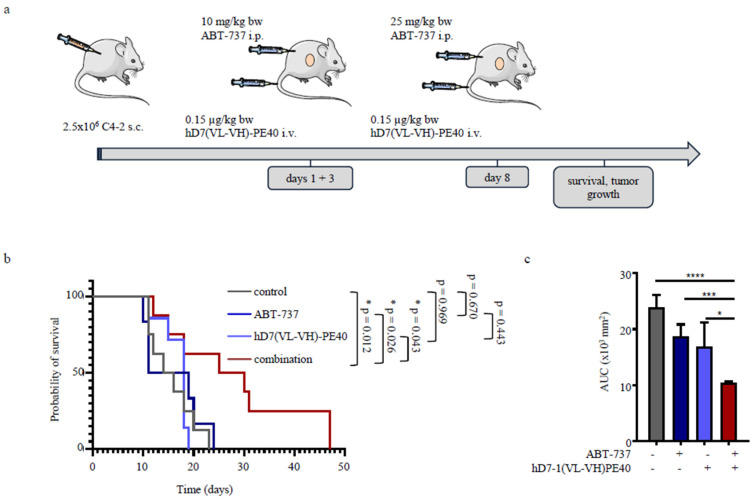
The combination therapy of ABT-737 plus hD7-1(VL-VH)-PE40 shows in vivo antitumor activity in the C4-2 SCID mouse xenograft model. (**a**) Schematic representation of animal treatment. (**b**) Animals treated with combination therapy showed significantly enhanced survival compared to the control groups. Kaplan-Meier analysis and log-rank test with * *p* < 0.05. (**c**) Animals treated with combination therapy showed significant inhibition of tumor growth compared to the control groups. Time-adjusted AUC with * *p* < 0.05; *** *p* < 0.001, **** *p* < 0.0001.

**Figure 7 cancers-12-01648-f007:**
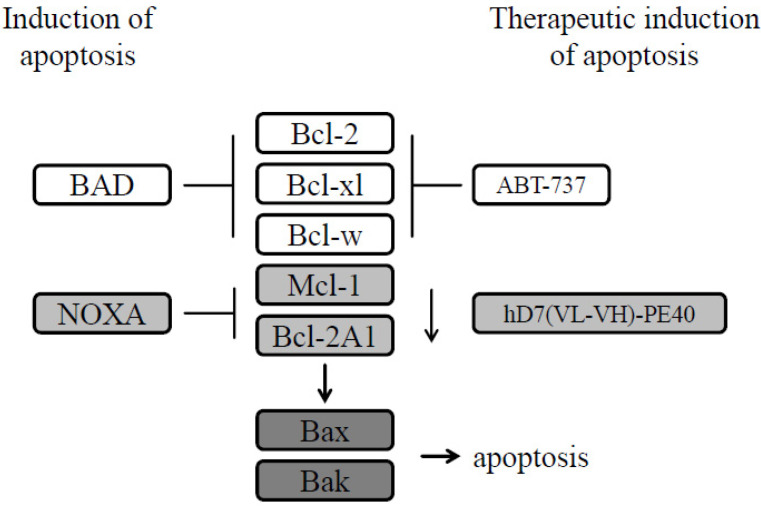
Schematic representation of the interactions between different members of the Bcl-2 protein family with ABT-737 and the immunotoxin hD7(VL-VH)-PE40 for the therapeutic induction of apoptosis. As a BAD-like BH3 mimetic ABT-737 sequesters Bcl-2, Bcl-xl, and Bcl-w, whereas the immunotoxin takes over the function of NOXA in that it reduces Mcl-1 and Bcl2A1 for the release of Bax and Bak.

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
