# Peer review of "An Anti-PSMA Immunotoxin Reduces Mcl-1 and Bcl2A1 and Specifically Induces in Combination with the BAD-Like BH3 Mimetic ABT-737 Apoptosis in Prostate Cancer Cells"

_cancers, 2020, doi:10.3390/cancers12061648_

Round 1
Reviewer 1 Report
The manuscript from Masilamani et al describes the synergistic action between ABT-737 and the immunotoxin hD7-1(VL-VH)-PE40 against PSMA-positive prostate cancer (PC) cells. This study extends a 2018 report by Philipp Wolf and co-workers that examined this synergy in vitro using a similar immunotoxin in conjunction with ABT-737. The current manuscript moderately extends this work by now using a humanized version of the immunotoxin and demonstrating its synergistic effects with ABT-737 both in vitro and in vivo. Overall, the conclusions reached in this manuscript are supported by the data presented. Although the subcutaneous tumor study using C4-2 cells provides important new data, a bone tumor study would have further enhanced the impact of the report. Minor concerns are listed below.
1) The Introduction should briefly discuss the value of PSMA as a potential therapeutic target for CRPC. It should also briefly mention the work of others targeting PSMA with antibodies in prostate cancer.
2) The statement (lines 78-79) “Bcl-2, Bcl-xl, and to a lesser extend Bcl-w, and are therefore also called BAD-like proteins” is confusing to this reviewer.
3) Add Figure 3 to line 150.
4) In Figure 5 “down spinning” should be replaced with spinning down.
5) Line 216 “gof” should be changed to of.
6) It would be helpful for some readers if the authors provide a brief explanation for the logic behind using the anti-puromycin antibody in Figure 2A and logic behind the Rho-123 assay used in Figure S2 to measure MOMP. One sentence each should be sufficient.
Author Response
We thank for the reviewing of the manuscript and for the helpful comments.
Changes in the manuscript are highlighted in a red font. Our point by point answers are as follows:
1) The Introduction should briefly discuss the value of PSMA as a potential therapeutic target for CRPC. It should also briefly mention the work of others targeting PSMA with antibodies in prostate cancer.
Brief information was inserted into the introduction about PSMA and anti-PSMA antibodies with corresponding references (lines 90-95).
2) The statement (lines 78-79) “Bcl-2, Bcl-xl, and to a lesser extend Bcl-w, and are therefore also called BAD-like proteins” is confusing to this reviewer.
The text has been corrected accordingly so that the statement is clearer (lines 79-81).
3) Add Figure 3 to line 150.
A reference to Figure 3 was added in the text (line 164).
4) In Figure 5 “down spinning” should be replaced with spinning down.
The text was changed in Figure 5.
5) Line 216 “gof” should be changed to of.
„Gof“ was changed to „of“ (line 235).
6) It would be helpful for some readers if the authors provide a brief explanation for the logic behind using the anti-puromycin antibody in Figure 2A and logic behind the Rho-123 assay used in Figure S2 to measure MOMP. One sentence each should be sufficient.
More information was included about puromycin Western Blot (lines 131-134) and Rho123 assay (lines 153-158) in the Results section.
Reviewer 2 Report
In the manuscript titled “An anti-PSMA immunotoxin reduces Mcl-1 and 3 Bcl2A1 and specifically induces in combination with 4 the BAD-like BH3 mimetic ABT-737 apoptosis in 5 prostate cancer cells” Masilamani and collaborators show that ABT-373, a molecule that target antiapoptotic proteins, but has shown to have minimal effects on Mcl1 and is associated to significant side effects, when in combination with an Pseudomonas exotoxin 1 toxic specifically guided to prostate cell via PSMA binding protein fragment exhibits potent tumor cytotoxic effects in in vivo and in vitro studies.
This is a well-reasoned and well written study that has significant clinical implication for future prostate cancer treatment.
I suggest two minor alterations
1) The X and Y axis in Figure 6b needs to be labeled.
2) The synergy of the two agents, while apparent can be better expressed using an appropriate equation such as the Bliss independence model equation.
Author Response
We thank for the reviewing of the manuscript and for the helpful comments.
Changes in the manuscript are highlighted in a red font. Our point by point answers are as follows:
1) The X and Y axis in Figure 6b needs to be labeled.
The axes were labeled in Figure 6.
2) The synergy of the two agents, while apparent can be better expressed using an appropriate equation such as the Bliss independence model equation.
We found synergism in combination with immunotoxin and ABT-737 on LNCaP and C4-2 cells after calculation with the Bliss independence model according to Zhao et al, 2014. This was added to the Statistics section (lines 417-419) and to the Results section (lines 177-185) of the manuscript.